# MicroRNA Sequencing Analysis in Obstructive Sleep Apnea and Depression: Anti-Oxidant and MAOA-Inhibiting Effects of miR-15b-5p and miR-92b-3p through Targeting PTGS1-NF-κB-SP1 Signaling

**DOI:** 10.3390/antiox10111854

**Published:** 2021-11-22

**Authors:** Yung-Che Chen, Po-Yuan Hsu, Mao-Chang Su, Ting-Wen Chen, Chang-Chun Hsiao, Chien-Hung Chin, Chia-Wei Liou, Po-Wen Wang, Ting-Ya Wang, Yong-Yong Lin, Chiu-Ping Lee, Meng-Chih Lin

**Affiliations:** 1Division of Pulmonary and Critical Care Medicine, Department of Medicine, Kaohsiung Chang Gung Memorial Hospital, Chang Gung University College of Medicine, Kaohsiung 83301, Taiwan; wanpasum@cgmh.org.tw (P.-Y.H.); maochangsu@yahoo.com.tw (M.-C.S.); cchsiao@mail.cgu.edu.tw (C.-C.H.); chestman@cgmh.org.tw (C.-H.C.); filling226@cgmh.org.tw (T.-Y.W.); yonyonlin@yahoo.com.tw (Y.-Y.L.); choupeen@cgmh.org.tw (C.-P.L.); 2Department of Medicine, College of Medicine, Chang Gung University, Taouyan 33302, Taiwan; 3Sleep Center, Kaohsiung Chang Gung Memorial Hospital and Chang Gung University College of Medicine, Kaohsiung 83301, Taiwan; 4Department of Respiratory Therapy, Chang Gung University of Science and Technology, Chia-yi 613016, Taiwan; 5Institute of Bioinformatics and Systems Biology, National Yang Ming Chiao Tung University, Hsinchu 30068, Taiwan; dodochen@nctu.edu.tw (T.-W.C.); r05445119@ntu.edu.tw (P.-W.W.); 6Department of Biological Science and Technology, National Yang Ming Chiao Tung University, Hsinchu 30068, Taiwan; 7Graduate Institute of Clinical Medical Sciences, College of Medicine, Chang Gung University, Taouyan 33302, Taiwan; 8Department of Neurology, Kaohsiung Chang Gung Memorial Hospital and Chang Gung University College of Medicine, Kaohsiung 83301, Taiwan; cwliou@ms22.hinet.net

**Keywords:** obstructive sleep apnea, microRNA, next generation sequencing, miR-92b, miR-15b, PTGS1, depression

## Abstract

The aim of this study was to identify novel microRNAs related to obstructive sleep apnea (OSA) characterized by intermittent hypoxia with re-oxygenation (IHR) injury. Illumina MiSeq was used to identify OSA-associated microRNAs, which were validated in an independent cohort. The interaction between candidate microRNA and target genes was detected in the human THP-1, HUVEC, and SH-SY5Y cell lines. Next-generation sequencing analysis identified 22 differentially expressed miRs (12 up-regulated and 10 down-regulated) in OSA patients. Enriched predicted target pathways included senescence, adherens junction, and AGE-RAGE/TNF-α/HIF-1α signaling. In the validation cohort, miR-92b-3p and miR-15b-5p gene expressions were decreased in OSA patients, and negatively correlated with an apnea hypopnea index. PTGS1 (COX1) gene expression was increased in OSA patients, especially in those with depression. Transfection with miR-15b-5p/miR-92b-3p mimic in vitro reversed IHR-induced early apoptosis, reactive oxygen species production, MAOA hyperactivity, and up-regulations of their predicted target genes, including PTGS1, ADRB1, GABRB2, GARG1, LEP, TNFSF13B, VEGFA, and CXCL5. The luciferase assay revealed the suppressed PTGS1 expression by miR-92b-3p. Down-regulated miR-15b-5p/miR-92b-3p in OSA patients could contribute to IHR-induced oxidative stress and MAOA hyperactivity through the eicosanoid inflammatory pathway via directly targeting PTGS1-NF-κB-SP1 signaling. Over-expression of the miR-15b-5p/miR-92b-3p may be a new therapeutic strategy for OSA-related depression.

## 1. Introduction

Affecting approximately 25% of men and 13% of women, obstructive sleep apnea (OSA) is characterized by recurrent episodes of upper airway collapse, which result in recurring arousals and desaturation during sleep, and lead to sleep fragmentation and chronic intermittent hypoxia with re-oxygenation (IHR) injury [1]. Significant adverse consequences of OSA syndrome include neurocognitive dysfunction (Alzheimer’s disease, depression), stroke, hypertension, heart failure, atrial fibrillation, diabetes mellitus, pulmonary hypertension, chronic kidney disease, and cancer. OSA was associated with the increased relative risks of all-cause mortality (pooled hazard ratio: 1.262) [2,3]. Continuous positive airway pressure (CPAP) is the mainstream therapy for OSA, but a recent meta-analysis of eight randomized controlled trials showed no evidence that CPAP therapy improves cardiovascular outcomes, suggesting that some detrimental pathogenesis continues despite treatment [4]. The mechanism underlying adverse consequences of OSA include dysregulation of hypoxia-inducible factor (HIF)-1/2 by chronic IHR feeding forward production of reactive oxygen species (ROS) in the carotid body chemosensory reflex pathway [5]. The imbalance between HIF-1α-dependent pro-oxidant and HIF-2α-dependent antioxidant enzymes promotes maladaptive responses to IHR and end-organ injury by enhancing pro-inflammatory pathways [6]. IHR exposure activates toll-like receptor/ nuclear factor kappa-light-chain-enhancer of activated B cells (NF-κB) p65 signaling, augmenting the expression of inflammatory cytokines and oxidative stress, and leading to atherosclerotic plaque growth and instability, and hippocampal neuronal damage [7,8]. Therefore, examining the molecular basis of adverse consequences in clinical populations with OSA and the potential effects on oxidative stress and inflammation can improve our understanding of the mechanistic underpinnings of OSA-related outcomes.

MicroRNAs (miRNAs) are small non-coding single-stranded RNAs of ~22 nucleotides in length and regulate up to 60% of the protein-encoding genome. The main mechanism with which microRNA affects protein-coding genes is by interaction with the 3′-untranslated region of target mRNA, subsequently leading to its degradation or translation repression [9]. MiRNAs participate in the proper development of the organism and the response to stress, including proliferation, apoptosis, cellular development, cellular signaling, and inflammation. A plethora of miRNAs has been shown to be up-regulated in response to IHR via directly targeting anti-inflammatory or antioxidant signaling, or their responsive genes, and possess a positive feedback loop to stabilize the HIF-1α protein, while the down-regulated miRNAs commonly suppress the expression of HIF-1α or pro-inflammatory signaling and engage in protective mechanisms against IHR injury. Current studies on miRNA in OSA are lacking in the validation of the findings in independent and diverse populations and lacking in consistency between expression changes in OSA patients and in response to IHR stimuli in vitro. Whole genome miRNA-sequencing allows the discovery of novel miRNAs, and the identification of biomarkers, predictors, and therapeutic targets of disease with high accuracy [10].

Here, we examined candidate OSA-related miRNAs in a two-step manner. First, differences in the expression of miRNAs in peripheral blood mononuclear cells (PBMCs) were identified by next-generation sequencing (NGS) between OSA patients and healthy non-snorers. Next, ten candidate miRNAs were selected for validation in an independent larger cohort of primary snoring (PS) subjects, treatment-native OSA patients, and long-term CPAP-treated OSA patients. Finally, functional effects of two validated miRNAs (miR-15b-5p and miR-92b-3p) on oxidative stress, inflammation, and monoamine oxidase A (MAOA) were examined by a series of in vitro IHR experiments, and their implication in the treatment of OSA-related depression was investigated. Since prostaglandin-endoperoxide synthase 1 (PTGS1; COX1), one of the predicted target genes of miR-15b-5p and miR-92b-3p, has been shown to modulate the nuclear factor kappa B (NF-κB) signaling pathway, which in turn augments MAOA expression via enhanced binding of Sp1 transcription factor (SP1) with GC box of the proximal MAOA gene promoter, we hypothesized that both miRNAs may regulate PTGS1 via NF-κB-SP1 signaling to modify depression caused by chronic IHR in OSA [11,12].

## 2. Materials and Methods

### 2.1. Participants

The study was approved by the Institutional Review Board of Chang Gung Memorial Hospital, Taiwan (certificate number: 201509756B0). The study participants were recruited from the sleep center and pulmonary clinics of Kaohsiung Chang Gung Memorial Hospital during the period from August 2016 through July 2019. Written informed consent was obtained from all subjects participating in the study, who were aged 20 years or older. Exclusion criteria included ongoing infections, any known autoimmune disease, recent use of an immunosuppressive agent in the past half-year, narcolepsy, morbid obesity (body mass index, BMI, >34 kg/m^2^), too old age (>65 years old), and too lean body weight (BMI < 21 kg/m^2^). A total of 94 participants were included in the study. Among them, 10 patients with severe OSA and 6 healthy non-snorers (defined as no OSA-related symptoms and no snoring history reported by bed partners) were enrolled as discovery cohort in the miRNA NGS experiments, while 20 subjects with primary snoring (PS, apnea-hypopnea index, AHI, <5), 45 treatment-naïve patients with severe OSA (AHI ≥ 30.0), and 13 long-term CPAP-treated severe OSA patients (AHI ≥ 30.0, CPAP use ≥4 h/night for ≥12 months) were enrolled as a validation cohort. Adverse consequences of OSA, including hypertension (baseline blood pressure > 140/90 mmHg), heart disease (including ischemic heart disease, cardiac dysrhythmia, and congestive heart failure), diabetes mellitus, stroke, depression (at least 1 of the 2 core symptoms of low mood and loss of interest, or taking antidepressant medication at Psychiatric Clinic), and chronic kidney disease were recorded.

### 2.2. Polysomnography and CPAP Titration Study

All the patients with sleep-disordered breathing underwent overnight polysomnography examination by using Sandman SD32+TM Digital Amplifier (Embla, Broomfield, CO, USA) at the sleep center of Kaohsiung Chang Gung Memorial Hospital. Sleep stage scoring was performed by trained technicians according to standard criteria [13]. The nocturnal hypoxic load was evaluated in terms of mean SaO2, nadir SaO2, the number of dips >4% of basal SaO2%//h (oxygen desaturation index, ODI), and the percentage of total minutes of recording time with SaO2 <90 % (%time <90% SaO2) [14]. A subset of OSA patients underwent CPAP titration study with a manually titrated machine (GoodKnight 420E, Nellcor Puritan Bennett, CA, USA) to get an optimal pressure before starting their treatment with auto-adjusted positive airway pressure machines at home. The Epworth Sleepiness Scale (ESS) recorded at the PSG exam was used to measure sleep propensity in every study participant [15].

### 2.3. Blood Collection and RNA Isolation

Fresh venous blood 20 mL was collected from each participant on awakening and immediately transferred to a tube containing 3.2% sodium citrate (1:10 dilution). PBMCs were isolated by Ficoll-Hypaque gradient centrifugation (HISTOPAQUE^®^-119, Sigma-Aldrich, Inc., Burlington, MA, USA), and then stored in RNAlater (Ambion Inc., Austin, TX, USA) at −80 °C until RNA isolation. A miRNeasy Mini Kit column-based system (Qiagen, Valencia, CA, USA) was used for isolation of total RNA, and treated with DNase. RNA samples were run on an RNA 6000 Nano Gel System (Agilent Technologies Inc., Santa Clara, CA, USA) using an Agilent 2100 Bioanalyzer (Agilent) to determine the quality of RNA. Only samples with A260/A280 ratios of 1.9 to 2.1 and RNA integrity number ≥ 8 were used for further analysis.

### 2.4. Whole Genome microRNA Profiles and Analysis by NGS

The pooled RNA libraries were subject to NGS assay to determine miRNA expression profiles. Illumina MiSeq platform was adopted for the profiling job. The RNA libraries were first prepared in accordance with the TruSeq Small RNA Sample Preparation protocol (Illumina) followed by sequencing with the MiSeq platform. Read counts for miRNAs were estimated with sRNAnalyzer for case and control miRNA sequencing datasets [16]. A total of 311 miRNAs with median read counts larger than 5 were used to identify differentially expressed miRNAs. The read counts were then normalized by Partek^®^ Genomics Suite^®^. Twenty-two miRNAs with absolute fold change larger than 2 or less than 0.5 and *p*-value less than 0.05 were identified as differentially expressed miRNAs with Partek^®^ Genomics Suite^®^. All the sequencing reads are publicly available on the NCBI website with an accession number of GSE174245.

### 2.5. Prediction of microRNA Target and Pathway Enrichment

Validated microRNA target databases, miRecords (v4) [17], miRTarBase (v7) [18] and TarBase (v8) [19] were queried by using R package multiMiR (v2.3.0) [20] to retrieve their targets in the format of Entrez gene ID. To obtain a gene list for pathway enrichment, we intersected the predicted target genes of 22 miRNAs and preserved the genes found in equal or more than three microRNAs, which resulted in 4238 genes. We further highlighted these genes in the pathway maps with IPA [21]. In over-representative analysis, the gene sets from the Gene Ontology database [22,23] and Kyoto Encyclopedia of Genes and Genomes (KEGG) database [24,25,26] were used. For the Gene Ontology database, the gene sets were acquired from R package org.Hs.eg.db (v3.12.0) and GO.db (v3.12.1). We only kept gene sets with gene sizes between 10 and 500. The hypergeometric tests were applied separately on the three ontology classes, biological process, molecular function, and cellular component. The *p*-values were corrected with Benjamini and Hochberg method [27] to control the false discovery rate. We defined the significant pathways with a false discovery rate-adjusted *p*-value less than 0.00001 for the biological process and 0.05 for both molecular function and cellular component. For each ontology class, we curated the significant pathways into multiple sub-categories and drew the pie chart. For the KEGG database, we use the R package clusterProfiler (v3.16.1) [28] to conduct enrichment analysis. Based on the current version of clusterProfiler, the latest gene sets were pulled from the KEGG database server at the time we performed the analysis (19 April 2021). The dynamic change of the KEGG database can be found on the website (https://www.kegg.jp/kegg/docs/statistics.html, accessed on 3 November 2021). The statistical tests were performed using the function enrich KEGG with default parameters.

### 2.6. Analysis of miRNAs by Quantitative Reverse-Transcriptase Polymerase Chain Reaction (RT-PCR) in an Independent Validation Cohort

cDNA was generated from 2 µL of purified total RNA using the TaqMan Advanced miRNA cDNA Synthesis kit (Thermo Fisher Scientific, Waltham, MA, USA). Additionally, 1 pM of the synthetic C. Elegans oligo, cel-miR-39 was added to the isolated total RNA. This sequence does not exist in humans and was used as an exogenous control. All qPCR reactions were normalized to their corresponding cel-miR-39 Ct values. Quantitative RT-PCR was performed for each sample using 2.5 µL of diluted cDNA, TaqMan Advanced miRNA Assays (Appendix A; Thermo Fisher Scientific, Waltham, MA, USA), and Applied Biosystems™ TaqMan™ Fast Advanced Master Mix (Thermo Fisher Scientific, Waltham, MA, USA) under fast cycling conditions. All TaqMan assays quantitative RT-PCR was carried out using the ABI 7500fast Real-Time PCR System (Applied Biosystems). Real-time PCR cycling conditions consisted of 95 °C for 20 s, followed by 40 cycles of 95 °C for 3 s and 60 °C for 30 s. All miRNA fold expression changes were determined by the 2^−ΔΔCT^ method.

### 2.7. Determination of Target Gene mRNA Expressions of Isolated PBMCs Using Quantitative RT-PCR

To determine the expressions of the predicted target genes, the gene expressions of the *TNFSF13B*, *VEGFA*, *AMOT*, *NOX4*, *CXCL5*, *LEP*, *PTGS1*, *ADRA2A*, *ADRB1*, *GABRA4*, *GABRB2*, *GABRB1*, *TNF-α*, *TGF-β1*, and *NF-κB1* genes were analyzed using quantitative RT-PCR in a 96-well format. The housekeeping gene GAPDH was chosen as an endogenous control to normalize the expression data for each gene. All PCR primers (random hexamers) were designed and purchased from Roche according to the company’s protocols (www.roche-applied-science.com, accessed on 13 June 2019), and their sequences are given in Appendix A. RNA samples were treated DNA-free to remove contaminating genomic DNA. A total of 300 ng RNA was used for the synthesis of first-strand cDNA with QuantiTectReverse Transcription Kit (QIAGEN, Germany). A total of 5 μL of the reverse transcription reaction was added to 5 μL of the master mix (QIAGEN, SYBR Green PCR kit; Roche, Germany). The PCR reactions with 45 cycles of amplification were run in a Roche LightCycle 480 machine. A single real-time PCR experiment was carried out on each sample for each target gene because the Roch Light CyclerQuantiFast R system has shown high reproducibility. Relative expression levels were calculated using the ΔΔCq method with the median value for the control group as the calibrator.

### 2.8. In Vitro IHR Stimuli in Cell Culture Models

Human monocytic cell line THP-1 cells obtained from ATCC (1 × 106 cells/mL) were suspended in a culture dish containing RPMI 1640 medium. Human umbilical vein endothelial cells (HUVECs) purchased from Lonza (Basel, Switzerland) were cultured in Clonetics Endothelial Cell Growth Media supplemented with the BulletKit containing bovine brain extract, epidermal growth factor, hydrocortisone, gentamicin, amphotericin B, 2% fetal bovine serum, 1% penicillin/streptomycin and ascorbic acid. Human SH-SY5Y neuron cells purchased from ATCC (^®^ CRL-2266™) were cultured in ATCC-formulated MEM/F12 (1:1) growth medium supplemented with 10% fetal bovine serum, 1x non-essential amino acids, 1 mM sodium pyruvate, 1.5 g/L sodium bicarbonate, and 1% penicillin/ streptomycin. Cells were exposed to IHR or normoxic conditions in custom-designed, incubation chambers which were attached to an external O_2_–CO_2_ hand-driven controller. IHR protocol consists of a 25-min hypoxic period (0% O_2_ and 5% CO_2_) and 35 min of the re-oxygenation period (21% O_2_ and 5% CO_2_) per cycle, 8 cycles /day for 3 days, which has been shown to achieve an episodic decrease in culture medium SaO2 by 30–40% [29].

### 2.9. Transfection with miRNA-15b-5p Mimic/miR-92b-3p Mimic

miR-15b-5p/miR-92b-3p mimic (final concentration, 10/25 nM) was synthesized by GenePharma, and was incubated in cells with Lipofectamine 2000 (Invitrogen, Carlsbad, CA, USA) for 6 h to over-express the gene expression level of miR-15b-5p/miR-92b-3p using the HiPerFect transfection reagent (QIAGEN, Hilden, Germany). The efficiency of the transfection was detected by quantitative RT-PCR.

### 2.10. Reporter Constructs, Mutagenesis and Luciferase Reporter Assay

For the luciferase reporter assay, we utilized pmirGLO Dual-Luciferase miRNA Target Expression Vector (pmirGLO) (Promega, Madison, WI, USA). Since there were miR-92b-3p binding sites in PTGS1 3′ un-translated region (3′ UTR), two plasmid constructs, PTGS1 wild type (miR-92b-3p binding site at position 3267 to 3273) and PTGS1 mutation type were created. For the luciferase reporter assay, we co-transfection the pmirGLO plasmids using the lipofectamine 3000 reagent (Thermo, Waltham, MA, USA) and different doses of miR-92b-3p mimic (0, 5, 10, 25, and 50 nM) using the HiPerFect transfection reagent (Qiagen, Hilden, Germany). The luciferase activity was measured using the Dual-Glo^®^ Luciferase Assay System (Promega, Madison, WI, USA).

### 2.11. Measurement of Cell Apoptosis by Flow Cytometry Analysis

Cell apoptosis rates were evaluated by flow cytometry using an Annexin V/Propidium iodide (PI) apoptosis detection kit (BD Biosciences, Franklin Lakes, NJ, USA). Following treatment, cells were washed twice with PBS, re-suspended in binding buffer, and incubated with 5 μL FITC-Annexin V and 5 μL PI for 15 min at room temperature. Staining cells were analyzed using the FACScan flow cytometry system (Becton Dickinson, San Diego, CA, USA).

### 2.12. Measurement of Intracellular Reactive Oxygen Species (ROS)

A fresh stock of 0.1 μM solution of H2DCFDA (catalog no. D6883; Sigma, USA) was added to the cells at a density of 1 × 10^6^ cells/mL. Cell-associated mean fluorescent intensity was measured by flow cytometry in the FL1 channel at excitation and emission wavelengths of 488 and 535 nm, respectively, using the CytomicsTM FC500 (Beckman Coulter, Brea, CA, USA).

### 2.13. Measurement of MAO Catalytic Activity

The MAO catalytic activity assay was used according to the manufacturer’s instructions (MAK136, Sigma, USA). Briefly, SH-SY5Y cells were lysated and incubated with clorgyline (MAO-A inhibitor), pargyline (MAO-B inhibitor), or both together (provided with the kit) in triplicate, for at least 10 min in the dark microplate to allow the inhibitor to interact with the enzyme. MAO assay mix (assay buffer, p-tyramine, HRP enzyme, and dye reagent) was then added to the samples and incubated the reaction for 20 min at room temperature. The microplate was read in a multimode microplate readers-fluorescence analyzer at room temperature with excitation/emission wavelengths of 530/585 nm, respectively.

### 2.14. Measurement of Cell Viability by WST-1

WST-1 reagent (Roche, Mannheim, Germany) diluted 1:10 in a growth medium was added into THP-1 cells grown in a 96-well plate (10^4^ cells/well) for the last 1 h according to the manufacturer’s instructions. The number of viable cells was determined via optical density measurement using a microplate reader at 450 nm, with 600 nm as a reference wavelength.

### 2.15. Immunofluorescence Stain

The preparation of cell cultures for immunofluorescence was performed using Millicell EZ 8-well glass slides (Merck Millipore) and SH-SY5Y cells were seeded at 5 × 104 on EZ 8-well glass slides. The cells were washed with phosphate-buffered saline (PBS) and fixed for 15 min at room temperature in 4% paraformaldehyde in PBS at pH 7.4. After removing the paraformaldehyde, the cells were rinsed three times in PBS and permeabilized for 10 min in 0.1% Triton X-100 in PBS. The permeation solution was removed and the cells were again washed three times with PBS, followed by blocked in 5% Fetal Bovine Serum (FBS) for 60 min at room temperature and incubated overnight at 4 °C with primary anti-COX1 antibodies (Abcam, ab109025) (1:25); anti-NF-kB1 antibodies (Sigma, HPA027305) (1:100); anti-SP1 antibodies (Millipore, 07-645) (1:250); anti-MAO-A antibodies (Abcam, ab126751) (1:50) and ROS (H2DCFDA, 2′,7′-Dichlorofluorescein diacetate) (Sigma, D6883) 0.1uM. The cells were then incubated for 1 h at room temperature with the secondary antibody of DyLightTM 488 Donkey anti-rabbit IgG (BioLegend, 406404). DAPI (Sigma, F6057) was used to label the nuclei. Fluorescence images were acquired using Olympus DP80. The images for each cell were counted under five randomly selected 200X fields using Image J software.

### 2.16. Statistical Analysis

Data were expressed as the mean ± standard deviation. Independent Student T-test or Mann–Whitney U test was used for comparing continuous values of two experimental groups, where appropriate. ANOVA test followed by post hoc analysis with Bonferroni test was used for comparing mean values of more than two experimental groups in case of normal and homogeneous data, while Brown-Forsythe test followed by post hoc analysis with Tamhane’s T2 test was used in case of normal and non-homogeneous data. Categorical variables were analyzed using the Chi-square test. Pearson’s correlation was used to determine the relationship between selected variables. Stepwise multiple linear regression analysis with all potential co-variables, including age, sex, BMI, AHI, ESS, history of smoking, history of alcoholism, and co-morbidities, entered in a single step was used to adjust *p*-values in the subgroup analyses. A *p*-value of less than 0.05 is considered statistically significant.

## 3. Results

### 3.1. 22 Differentially Expressed miRs in OSA Patients Versus Healthy Non-Snorers in the NGS Discovery Experiment

Demographic, sleep, and biochemistry data of the discovery cohort are shown in Table 1. There was no difference between case and control groups in terms of age, gender, smoking history, alcoholism history, BMI, and co-morbidity. Whole genome NGS analysis and heatmap clustering (Figure 1A) identified 22 differentially expressed miRNAs in 16 treatment-naïve OSA patients versus eight healthy non-snorers (12 up-regulated, Figure 2A–L: miR-10a-5p (MIMAT0000253), miR-16-1-5p (MIMAT0000069), miR-18a-5p (MIMAT0000072), miR-106a-5p (MIMAT0000103), miR-146b-5p (MIMAT0002809), miR-148b-3p (MIMAT0000759), miR-223-5p (MIMAT0004570), miR-335-5p (MIMAT0000765), miR-374b-5p (MIMAT0004955), miR-421-3p (MIMAT0003339), miR-let-7a-1-3p (MIMAT0004481), and miR-let-7a-1-5p (MIMAT0000062); and 10 down-regulated, Figure 3A–J: miR-15b-5p (MIMAT0000417), miR-133a-1-3p (MIMAT0000427), miR-145-5p (MIMAT0000437), miR-150-5p (MIMAT0000451), miR-26b-3p (MIMAT0004500), miR-29c-5p (MIMAT0004673), miR-326-3p (MIMAT0000756), miR-4433b-3p (MIMAT0030414), miR-574-3p (MIMAT0003239), miR-92b-3p (MIMAT0003218); all fold changes >2 or <0.5, transcript per million >1000, all *p*-values < 0.05). We used several computational databases for the target predictions of the 22 miRNAs and identified 1996 individual genes. To evaluate the biological role of the differentially expressed miRNA target genes, we performed a Gene Ontology (GO) classification enrichment analysis (Figure 1B–D). Enriched predicted target pathways of the 22 candidate miRNA genes included cell senescence, p53, Estrogen Receptor Signaling adherens junction, MSP-RON signaling in cancer cells pathway, HGF Signaling, and HOTAIR signaling (Appendix A). We also performed (KEGG) pathways enrichment analysis for differentially expressed miRNA target genes. The significant KEGG pathways and their genes are shown in Table 2. Here, we highlight some of the pathways with potential biological significance in OSA, such as cellular senescence, adherens junction, AGE-RAGE signaling pathway in diabetic complications, TNF-α signaling pathway, insulin resistance, and HIF-1 signaling pathway. Based on potential biological significance, we further constructed gene networks for two pathways: (1) cell senescence (Figure 4) and (2) HIF-1α signaling (Appendix A).

### 3.2. Down-Regulated miR-15b-5p/miR-92b-3p in Treatment-Naïve OSA Patients Versus either PS Subjects or OSA Patient on CPAP Treatment in the Validation Cohort

Ten candidate miRs with potential biological or functional relevance, including miR-335-5p, miR-148b-3p, miR-223-5p, miR-16-1-5p, miR-let-7a-1-5p, miR-4433b-3p, miR-15b-5p, miR-92b-3p, miR-133a-1-3p, and miR-145-5p, were selected for further verification and validation. Demographic, sleep, and biochemistry data of the validation cohort are shown in Table 1. There was no difference between case and control groups in terms of age, gender, smoking history, alcoholism history, BMI, co-morbidity, and blood cholesterol/triglyceride/glycohemoglobin (HbA1c) levels, but the proportion of hypertension was higher in the OSA on CPAP group and there were significant differences in sleep parameters, such as AHI, ODI, and nadir SaO2. In the validation cohort, miR-15b-5p expression was decreased in treatment-naïve OSA patients (3.1 ± 5.8-fold change, Figure 5A) versus either PS (54.9 ± 95.4 fold change, adjusted *p* = 0.001) or OSA on CPAP (21.1 ± 29.2 fold change, adjusted *p* < 0.001) group. In treatment-naïve OSA patients and PS subjects, miR-15b-5p expression was negatively correlated with AHI (R = −0.346, *p* = 0.005, Figure 5B), ODI (R = −0.278, *p* = 0.025, Figure 5C), and arousal index (R = −0.269, *p* = 0.03), and positively correlated with nadir SaO2 (R = 0.246, *p* = 0.048) and mean SaO2 (R = 0.261, *p* = 0.036). Likewise, miR-92b-3p expression was decreased in treatment-naïve OSA patients (2.8 ± 8.2-fold change, Figure 5D) versus either PS (43.8 ± 97.7 fold change, adjusted *p* = 0.004) or OSA on CPAP (15.4 ± 21.1 fold change, adjusted *p* < 0.001) group. In treatment-naïve OSA patients and PS subjects, miR-92b-3p expression was negatively correlated with AHI (R = −0.289, *p* = 0.02, Figure 5E), and positively correlated with mean SaO2 (R = 0.248, *p* = 0.047, Figure 5F).

### 3.3. Up-Regulated PTGS1 in Treatment-Naive OSA Patients and Depression in the Validation Cohort

To determine the target genes related to miR-15b-5p and miR-92b-3p, the common targets and pathways of both miRs regulated by hypoxia were explored by the genes intersection option using the ingenuity pathway analysis (IPA) and miRbase database. The results identified several miR-15b-5p-regulated targets (three direct targets: *TNFSF13B*, *VEGFA*, *AMOT*), miR-92b-3p-regulated targets (two direct: *NOX4*, *CXCL5*), and common targets and pathways (Appendix A) regulated by both miRs (seven direct: *LEP*, *PTGS1*, *ADRA2A*, *ADRB1*, *GABRA4*, *GABRB2*, *GABRB1*; three indirect targets: *TNF-α*, *TGF-β1*, *NF-κB1*). Some of them were involved in pro-inflammatory, angiogenesis, or pro-oxidant responses and thus selected for further evaluation in both clinical samples and in vitro experiments. In the validation cohort, *PTGS1* (*COX1*) gene expression was increased in treatment-naïve OSA (18.1 ± 30.7 fold change, Figure 5G) versus either PS (1.6 ± 2.3 fold change, adjusted *p* = 0.007) or OSA on CPAP (5.4 ± 5.9-fold change, adjusted *p* = 0.024) group. *PTGS1* gene expression (24.2 ± 41.3 versus 3.3 ± 3.2-fold change, adjusted *p* = 0.002, Figure 5H) was increased in all sleep-disordered breathing patients with depression versus those without depression. *GABRB2* (R = 0.307, *p* = 0.003, Figure 5I) and *ADRB1* (R = 0.753, *p* < 0.001, Appendix A) gene expressions were positively correlated with HbA1C, and the latter was negatively correlated with nadir SaO2 (R = −0.432, *p* < 0.001, Appendix A). *TNFSF13B* gene expression was positively correlated with snoring index (R = 0.309, *p* = 0.003, Appendix A). *LEP* gene expression was negatively correlated with nadir SaO2 (R = −0.327, *p* = 0.001, Appendix A), and positively correlated with %TSaO2 < 90% (R= 0.498, *p* < 0.001, Appendix A) and HbA1C (R = 0.641, *p* < 0.001, Appendix A).

### 3.4. MiR-15b-5p Over-Expression Reversed IHR-Induced Apoptosis, ROS Production, and Target Gene Up-Regulations

IHR treatment in vitro resulted in down-regulation of both the miR-92b-3p and miR-15b-5p genes, while miR-15b/5p and miR-92b-3p transfection resulted in efficient over-expression of both miRs (Appendix A). Transfection with miR-15b-5p mimic at 25 nM in THP-1 cells reversed IHR-induced early apoptosis (percentage of Annexin V(+) PI (−) cells, Figure 6A), and IHR-induced up-regulation of its predicted target genes, including *PTGS1* (Figure 6B), *GABRB2*, *ADRB1*, *LEP*, *GARG1*, *TNFSF13B*, and *VEGFA* (Appendix A). Transfection with a miR-15b-5p mimic in HUVEC reversed IHR-induced ROS production (H2DCFDA incorporation, MFI, Figure 6C), IHR-induced early apoptosis (percentage of Annexin V(+) PI (−) cells, Figure 6D), and IHR-induced up-regulation of the *NF-κB1*, *TNF-α*, and *TGF-β* genes (Figure 6E–G). Transfection with a miR-15b-5p mimic in SH-SY5Y cells reversed IHR-induced ROS production (percentage of H2DCFDA, Figure 6H), late apoptosis (percentage of PI(+) Annexin V(+) cells, Figure 6I).

### 3.5. MiR-92b-3p Over-Expression Reversed IHR-Induced Apoptosis, ROS Production, and Target Gene Up-Regulations

Transfection with miR-92b-3p mimic at 25 nM in THP-1 cells reversed IHR-induced early apoptosis (Annexin V protein expression, MFI, Figure 7A) and up-regulations of the *CXCL5* and *ADRB1* genes (Appendix A. Transfection with a miR-92b-3p mimic in HUVEC reversed IHR-induced up-regulation of the *NF-κB1* (Figure 7B), *PTGS1* (Figure 7C), *TNF-α*, and *TGF-β* (Appendix A) genes. Transfection with a miR-92b-3p mimic in SH-SY5Y cells reversed IHR-induced ROS production (percentage of H2DCFDA, Figure 7D), early apoptosis (Annexin V expression, MFI, Figure 7E), and MAOA hyperactivity (percentage of normoxic condition, Figure 7F), while MAOB activity was not altered.

### 3.6. MiR-92b-3p Negatively Regulated PTGS1 in a Direct Manner

Analyses above suggested a negative correlation between miR-92b-3p expression and *PTGS1* expression. Hence, we assumed that miR-92b-3p might be directly targeting PTGS1, and this assumption was supported by a bioinformatics analysis using the TargetScan software. Analysis based on the dual-luciferase reporter assays further revealed that the pmirGLO luciferase activity altered in the wild-type *PTGS1* 3′-untranslated regions but remained unchanged in the mutant site of the *PTGS1* 3′-untranslated region (Figure 7G–H), which indicates that it was the target binding site for miR-92b-3p.

### 3.7. Knockdown of PTGS1 or Overexpression of miR-15b-5p/miR-92b-3p Alleviates IHR-Induced Neuron Cell Injury, Oxidative Stress, and MAOA Hyperactivity via Mediating NF-κB1-SP1 Signaling

Finally, we tried to validate whether PTGS1 is linked to the protective effects of miR-15b-5p/miR-92b-3p on IHR-induced injury. SH-SY5Y cells were transfected with Si-*PTGS1*, miR-15b-5p/miR-92b-3p mimic, and negative control. IHR treatment resulted in down-regulation of the miR-15b-5p (Figure 8A) and miR-92b-3p (Figure 9A) genes and up-regulation of the *PTGS1* gene (Figure 8D and Figure 9D) in SH-SY5Y cells, while transfection with miR-15b-5p mimic, miR-92b-3p mimic, and *PTGS1* SiRNA resulted in efficient over-expression or knock-down of the genes. Either PTGS1 knock-down or miR-15b-5p/miR-92b-3p over-expression at a concentration of 25 nM reversed IHR-induced cell viability decrease (WST1 percentage of normoxic condition, Figure 8B and Figure 9B), MAOA hyperactivity (Figure 8C and Figure 9C), and *SP1*/*NF-κB1* up-regulation (Figure 8E–F and Figure 9E–F), while MAOB activity was not altered. Immunofluorescence staining results further confirmed that IHR resulted in over-expressions of *PTGS1* (Figure 10A)/*NF-kB1*/*SP1* (Appendix A), MAOA hyperactivity (Figure 10B), and over-production of ROS (Appendix A), all of which were significantly reversed with the knock-down of *PTGS1*.

## 4. Discussion

In this study, we identified differentially expressed miRNAs associated with OSA on a genome-wide scale and validated two miRNAs along with corresponding changes in their target genes in an independent cohort. The 22 differentially expressed miRs regulate important genomic pathways, such as cellular senescence, cell cycle, and adherens junction, while the validated miR-15b and miR-92b regulate pro-inflammatory, pro-oxidant, adrenergic signaling, and GABAergic signaling. The present study demonstrated that miR-15b-5p and miR-92b-3p were down-regulated in both the treatment-naïve OSA patients and the IHR-exposed THP-1/HUVEC/SH-SY5Y cell models, while both miR-15b-5p mimic and miR-92b-3p mimic reversed IHR-induced ROS production, apoptosis, MAOA hyperactivity, and up-regulations of their target genes. In addition, the inverse expression patterns of miR-92b-3p and PTGS1 suggest their direct interaction, which has been verified through dual-luciferase reporter assays. Furthermore, either miR-15b-5p/miR-92b-3p over-expression or *PTGS1* knock-down reversed IHR-induced up-regulations of the *PTGS1/NF-κB1/SP1* genes. Thus, miR-92b-3p might play anti-inflammatory, antioxidant, and MAOA-inhibiting roles in the progression of OSA and the development of depression by regulating PTGS1 via NF-κB1/SP1 signaling.

Among the 22 differentially expressed miRNAs identified by NGS, miR-106a-5p, miR-574-3p, and miR-145-5p have been shown to be dysregulated in OSA patients in previous studies, while miR-26b-3p, miR-15b-5p, miR-16-5p, miR-29c-5p, miR-145-5p, miR-133a, and miR-223 gene expressions found to be skewed in response to IHR stimuli in vitro or in animal models [30,31,32]. In accordance with our findings, miR-145-5p has been demonstrated to be down-regulated in OSA patients and protect animals from aortic remodeling through targeting Smad3 in chronic intermittent hypoxic canine models [33,34]. Pathway analysis demonstrated that the 22 differentially expressed miRNAs are related to distinct molecular pathways associated with cellular senescence, cell cycle, adherens, tight junction, atherosclerosis, TGF-β signaling, TNF-α signaling, insulin resistance, Alzheimer’s disease, and HIF-1α signaling, all of which have been found to play a role in the development of adverse consequences of OSA. Accordingly, recent studies have shown that OSA may exacerbate vascular senescence via oxidative stress-related pathways through exosomes, accelerate chromosomal aging as evidenced by shortened telomere length, and trigger a senescence-like phenotype in pre-adipocytes [35,36,37,38]. Abnormal activity of the core cell-cycle machinery represents a driving force of tumorigenesis, while OSA has been recognized as a risk factor for cancer growth and aggressiveness mainly through HIF-1α signaling, which controls the synthesis of molecules with effects on inflammation, immune surveillance, and cell proliferation [39]. OSA is independently associated with impaired endothelial function and atherosclerosis through inflammation, oxidative stress, autonomic nervous system activation, and platelet activation [40,41]. Specifically, up-regulations of the miR146b, miR-421, miR-10a, miR-106a, miR-18a, miR-374b, miR-223, and miR-335 genes are implicated in the progression of atherosclerosis, cancer, oxidative stress, ischemic stroke, or insulin resistance, while down-regulations of the miR-150, miR-29c, miR-133a, and miR-145 genes are implicated in protection from heart failure, cancer, or insulin resistance [6,42,43].

In the current study, miR-15b-5p and miR-92b-3p were down-regulated both in the two independent cohorts of OSA patients and in response to IHR stimuli in vitro, and able to counteract oxidative stress-related cell apoptosis. In line with our findings, miR-15b has been shown to inhibit angiogenesis in proliferative diabetic retinopathy via targeting VEGFA, inhibit vascular smooth muscle cells in peripheral artery disease via targeting IGF1R, counteract senescence-associated mitochondrial dysfunction in skin aging via targeting SIRT4, and suppress Th17 Differentiation in multiple sclerosis by targeting O-GlcNAc [42,43,44,45]. In contrast, miR-15b has been found to augment cell apoptosis in Parkinson’s disease via targeting the GSK-3β/β-catenin signaling pathway, contribute to depression-like behavior in mice by affecting synaptic protein levels and function in the nucleus accumbens, deteriorate cardiomyocyte apoptosis in myocardial infarction via targeting Bcl-2/MAPK3, and contribute to extra-cellular matrix degradation in intervertebral disc degeneration via targeting SMAD3 [46,47,48,49]. In accordance with our findings, miR-92b-3p can suppress angiotensin II-induced cardiomyocyte hypertrophy via targeting HAND2, protect astrocyte neurons against oxygen and glucose deprivation, promote spinal cord neurite growth and functional recovery through the PTEN/AKT pathway [50,51,52]. In contrast, miR-92b can promote the progression of liver fibrosis by activating JAK/STAT pathway via targeting CREB3L2 and limiting the production of intermediate cortical progenitors [53,54]. The role of miR-92b and miR-15b in IHR injury characterized by OSA remains unclear. Our results revealed that miR-15b-5p/miR-92b-3p mimic can inhibit ROS production, MAOA activity, and apoptosis in vitro. Here, from searching the miRBASE web, we found PTGS1 may be a direct downstream target gene of miR-92b-3p. The dual-luciferase report gene assay determined that PTGS1 was the functional target of miR-92b-3p in HUVEC, and the gain-of-function/loss-of-function and immunofluorescence stain evaluation in SH-SY5Y cells confirmed the targeted relationship between miR-15b-5p/miR-92b-3p and PTGS1 through the NF-κB1-SP1 signaling pathway.

The pooled prevalence of depression in OSA patients is 35%, and OSA patients are at twice the risk of developing depression during follow-ups than those without OSA [55,56]. Elevated MAOA activity has been pointed out as a mechanism implicated in depression through producing ROS and catalyzing levels of all the three major monoamines (serotonin, norepinephrine, and dopamine) in the brain [57]. Irreversible MAO inhibitors have the potential to treat resistant depression, atypical depression, and bipolar depression, but are often reserved as back-line medicines because of their side effects [58]. Increased enzyme activity of PTGS1 and PTGS2 has been implicated as another mechanism in depression, and herb drugs reducing arachidonic acid levels through inhibiting PTGS1/2 could be used to treat depression in the chronic unpredictable mild stress rat model [59]. Previous studies have shown that miRNAs participate in a series of important pathophysiological processes of depression, but none have been investigated in OSA-related depression [60]. For the first time, we found that miR-15b-5p/miR-92b-3p mimics could inhibit MAOA activity through directly targeting PTGS1 via NF-κB1/SP1 signaling and may be developed as new drug targets for depression treatment in OSA patients. Our study examined the previously untouched area of the effect of miR-15b/miR-92b on OSA-related depression in vivo and in vitro. Clinical data showed that both miR-15b-5p and miR-92b-3p were down-regulated in OSA patients, while their common target gene, PTGS1, was up-regulated, particularly in those with depression. In vitro experiments showed that miR-15b/miR-92b targeting of PTGS1-NF-κB1-SP1 signaling selectively inhibits MAOA activity. To validate the result, we confirmed a reduction of MAOA activity and ROS production by using the immunofluorescence method with the knock-down of PTGS1 under normoxic and IHR conditions. Our findings indicate that miR-15b-5p/miR-92b-3p mimics may alleviate the neuronal damage and MAOA hyperactivity caused by chronic IHR via inhibiting neuroinflammation and oxidative stress. Since this report is an initial study on the role of miR-15b/miR-92b in MAOA activity for the development of OSA-related depression, a limitation should be acknowledged. The underlying mechanism of PTGS1, the miR-15b/miR-92b common targeting protein, involved in the pathogenesis of depression, is still ambiguous. Notwithstanding that limitation, this study does correlate miR-15b/miR-92b under-expression and PTGS1 over-expression with MAOA hyperactivity, oxidative stress, and augmented apoptosis of neuron cells in OSA-related depression. Our study sheds considerable light on the direction for developing a therapeutic approach for alleviating OSA-related depression and provides a potential target.

## 5. Conclusions

Our NGS experiment identified a selective cluster of miRNAs that regulates relevant biological pathways underlying disease severity and adverse consequences in OSA. Next, we validated that miR-15b-5p/miR-92b-3p and their target mRNA, PTGS1, were substantially decreased and increased, respectively, in OSA both in vivo and in vitro, while PTGS1 were further increased in OSA patients with depression and in response to IHR stimuli. Moreover, we demonstrated a critical role of miR-15b-5p/miR-92b-3p in apoptosis, oxidative stress, and MAOA activity of neurons by regulating the PTGS1-mediated eicosanoid inflammatory pathway via NF-κB1-SP1 signaling (Figure 11A,B). Hence, targeting the MAOA-related PTGS1 signaling pathway by miR-15b-5p/miR-92b-3p could provide a novel therapeutic avenue for treating OSA-related depression.

## Figures and Tables

**Figure 1 antioxidants-10-01854-f001:**
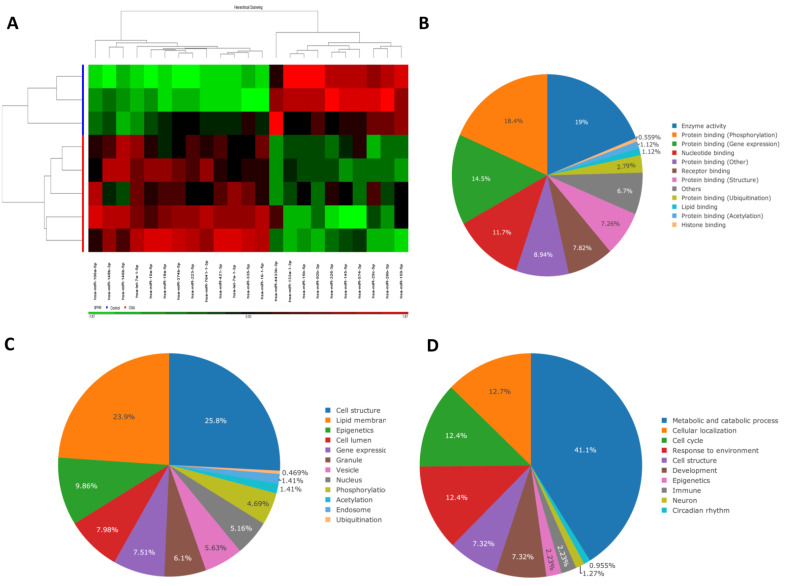
Peripheral blood mononuclear cell (PBMC)-derived microRNA (miRNA) profiling and Gene Ontology (GO) analysis. (**A**) Heatmap illustrating miRNA expression profiling between 16 OSA patients and eight healthy non-snorers (red: increased miRNA expression; green: reduced miRNA expression). GO analysis was used to assess (**B**) molecular functions, (**C**) cellular components, and (**D**) biological processes. The classification shows the top-ranked significant GO terms.

**Figure 2 antioxidants-10-01854-f002:**
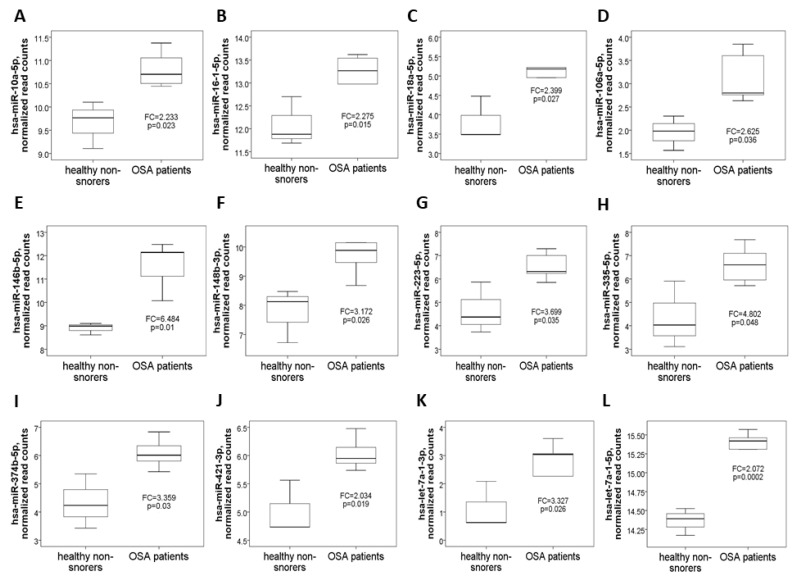
Twelve up-regulated microRNAs identified by next-generation sequencing in the comparison between OSA patients and healthy non-snorers in the discovery cohort. Normalized read counts of the (**A**) miR-10a-5p, (**B**) miR-16-1-5p, (**C**) miR-18a-5p, (**D**) miR-106a-5p, (**E**) miR-146b-5p, (**F**) miR-148b-3p, (**G**) miR-223-5p, (**H**) miR-335-5p, (**I**) miR-374b-5p, (**J**) miR-421-3p, (**K**) miR-let-7a-1-3p, and (**L**) miR-let-7a-1-5p genes were increased in OSA patients versus healthy non-snorers.

**Figure 3 antioxidants-10-01854-f003:**
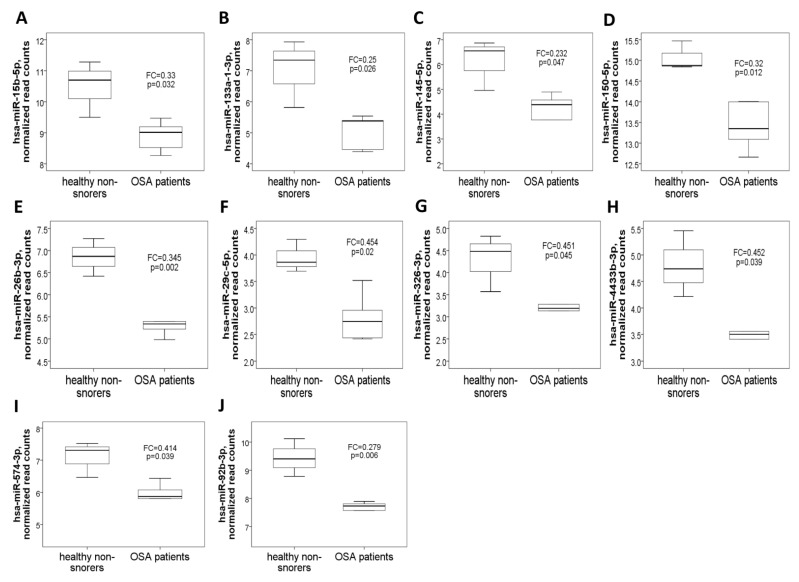
Ten down-regulated microRNAs identified by next-generation sequencing in the comparison between OSA patients and healthy non-snorers in the discovery cohort. Normalized read counts of the (**A**) miR-15b-5p, (**B**) miR-133a-1-3p, (**C**) miR-145-5p, (**D**) miR-150-5p, (**E**) miR-26b-3p, (**F**) miR-29c-5p, (**G**) miR-326-3p, (**H**) miR-4433b-3p, (**I**) miR-574-3p, and (**J**) miR-92b-3p genes were decreased in OSA patients versus healthy non-snorers.

**Figure 4 antioxidants-10-01854-f004:**
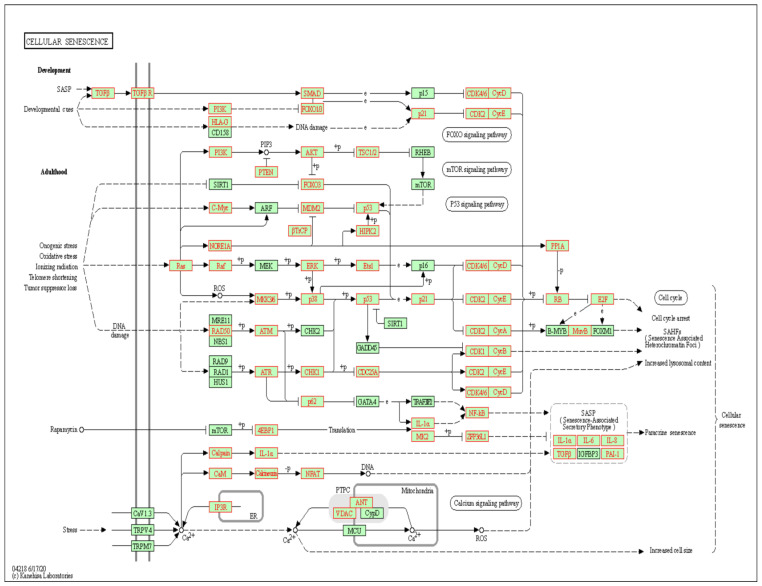
Gene pathway representation of interactions between gene target predictions of the 22 differentially expressed microRNAs. The most significant KEGG pathway of cellular senescence is identified by over-representation analysis with the gene list derived from the validated target genes of the 22 miRNAs (see Section 2). The FDR-adjusted *p*-value of the pathway is 1.2 × 10^−13^. Predicted target genes are shown in red letters and boxes with red lines.

**Figure 5 antioxidants-10-01854-f005:**
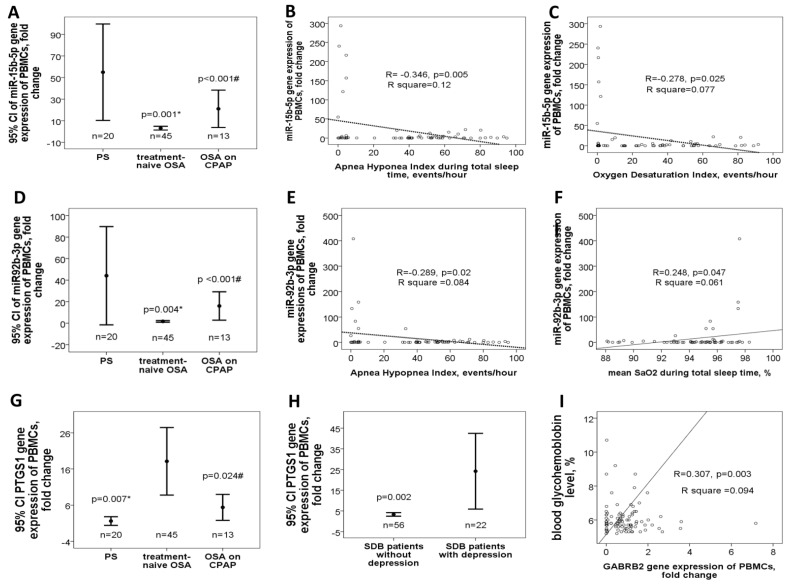
Down-regulated miR-15b-5p/miR-92b-3p genes and corresponding changes of their predicted target genes in the validation cohort. MiR-15b-5p gene expression was (**A**) decreased in treatment-naïve OSA patients versus either PS subjects or OSA patients on long-term CPAP treatment, and negatively correlated with (**B**) apnea hypopnea index (AHI) and (**C**) oxygen desaturation index. miR-92b-3p gene expression was (**D**) decreased in treatment-naïve OSA patients versus either PS subjects or OSA patients on long-term CPAP treatment, (**E**) negatively correlated with AHI, and (**F**) positively correlated with mean oxygen saturation (SaO2) during total sleep time. PTGS1 gene expression was (**G**) increased in treatment-naïve OSA patients, and (**H**) further increased in those with depression. (**I**) ADRB1 gene expression was negatively correlated with nadir SaO2.

**Figure 6 antioxidants-10-01854-f006:**
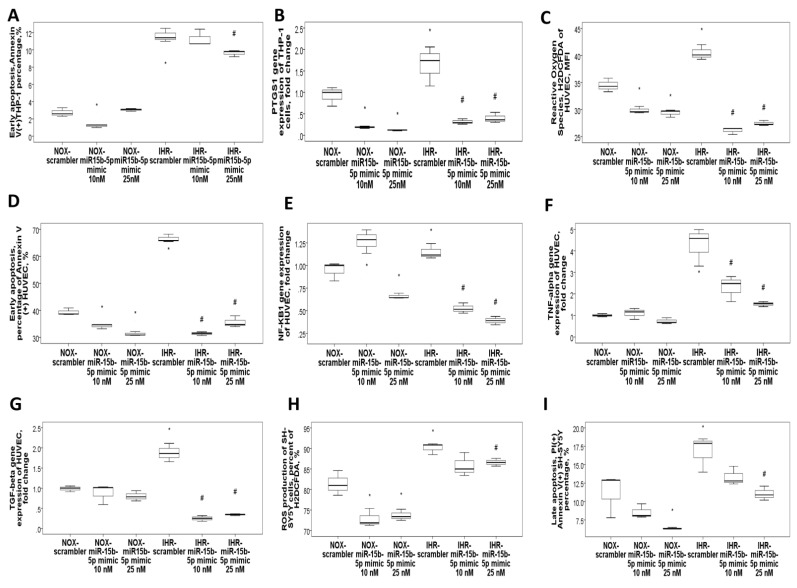
MiR-15b-5p over-expression reversed intermittent hypoxia with re-oxygenation (IHR)-induced apoptosis, oxidative stress, and up-regulation of its target genes. Transfection with a miR-15b-5p mimic in THP-1 cells reversed IHR-induced (**A**) early apoptosis and reversed IHR-induced up-regulation of its predicted target mRNA, (**B**) *PTGS1*. Transfection with miR-15b-5p in HUVEC reversed IHR-induced (**C**) reactive oxygen species production, (**D**) early apoptosis, and up-regulation of its predicted target genes, including (**E**) *NF-κB1*, (**F**) *TNF-α*, and (**G**) *TGF-β*. Transfection with a miR-15b-5p mimic in SH-SY5Y cells reversed IHR-induced (**H**) ROS production and (**I**) late apoptosis. * *p* < 0.05, compared with normoxic (NOX) condition. # *p* < 0.05, compared with IHR condition.

**Figure 7 antioxidants-10-01854-f007:**
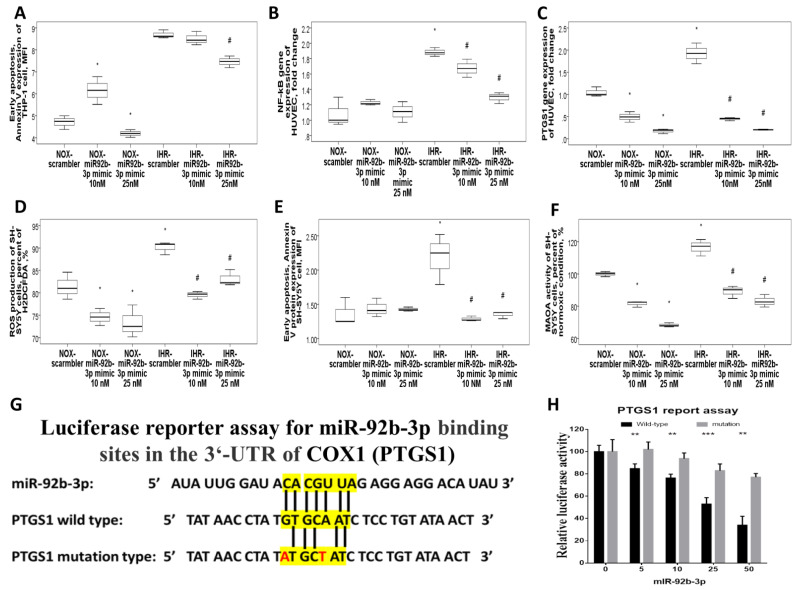
MiR-92b-3p over-expression reversed intermittent hypoxia with re-oxygenation (IHR)-induced apoptosis, monoamine oxidase (MAO)-A hyperactivity, and up-regulation of its target mRNAs through directly targeting PTGS1. Transfection with a miR-92b-3p mimic in THP-1 cells reversed IHR-induced (**A**) early apoptosis. Transfection with miR-92b-3p in HUVEC reversed IHR-induced up-regulation of its target genes, including (**B**) *NF-κB1*, and (**C**) *PTGS1*. Transfection with miR-92b-3p in SH-SY5Y cells reversed IHR-induced (**D**) ROS production, (**E**) early apoptosis, and (**F**) MAOA hyperactivity. (**G**) The wild-type sequence and mutated sequence of putative miR-92b-3p binding sites in the 3′untranslated region of PTGS1 are shown. (**H**) The direct binding between miR-92b-3p and *PTGS1* was confirmed using luciferase reporter gene assays. * *p* < 0.05, compared with normoxic (NOX) condition. ** *p* < 0.01, compared with normoxic (NOX) condition. *** *p* < 0.001, compared with NOX condition. # *p* < 0.05, compared with IHR condition.

**Figure 8 antioxidants-10-01854-f008:**
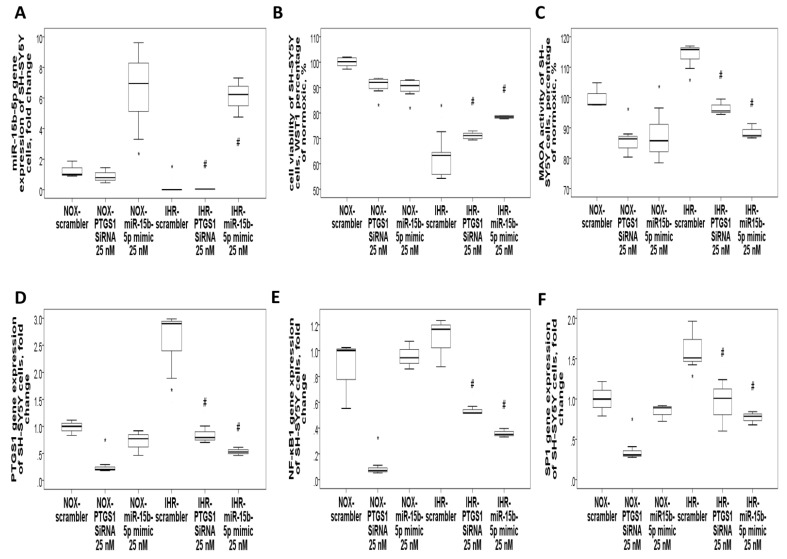
Over-expression of miR-15b-5p or knockdown of PTGS1 reversed intermittent hypoxia with re-oxygenation (IHR)-induced cell injury and MAOA hyperactivity via targeting NF-κB1/SP1 signaling. Transfection with either *PTGS1* SiRNA or miR-15b-5p mimic in SH-SY5Y neuron cells reversed IHR-induced (**A**) down-regulation of miR-15b-5p, (**B**) cell injury, (**C**) MAOA hyperactivity, and up-regulations of (**D**) *PTGS1*, (**E**) *NF-κB1*, and (**F**) *SP1*. * *p* < 0.05 compared with normoxic (NOX) condition. # *p* < 0.05 compared with IHR condition.

**Figure 9 antioxidants-10-01854-f009:**
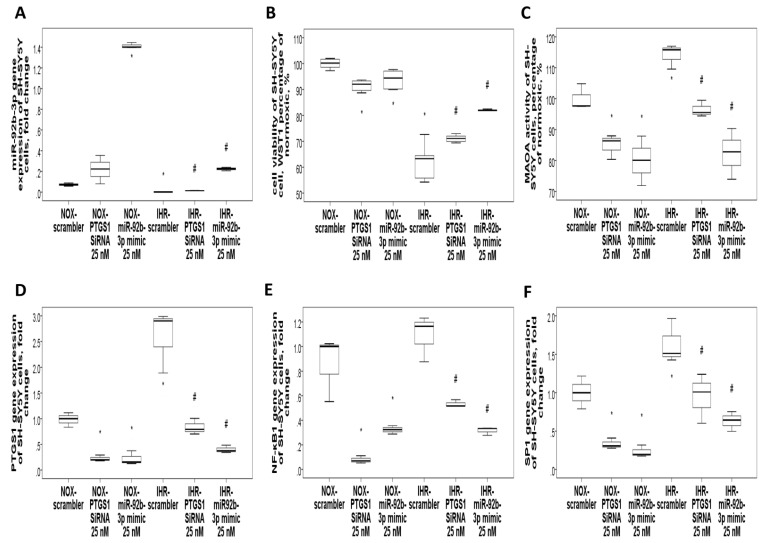
Over-expression of miR-92b-3p or knockdown of PTGS1 reversed intermittent hypoxia with re-oxygenation (IHR)-induced cell injury and MAOA hyperactivity via targeting NF-κB1/SP1 signaling. Transfection with either PTGS1 SiRNA or miR-92b-3p mimic in SH-SY5Y cells reversed IHR-induced (**A**) down-regulation of miR-92b-3p, (**B**) cell injury, (**C**) MAOA hyperactivity, and up-regulations of (**D**) PTGS1, (**E**) NF-κB1, and (**F**) SP1. * *p* < 0.05 compared with normoxic (NOX) condition. # *p* < 0.05 compared with IHR condition.

**Figure 10 antioxidants-10-01854-f010:**
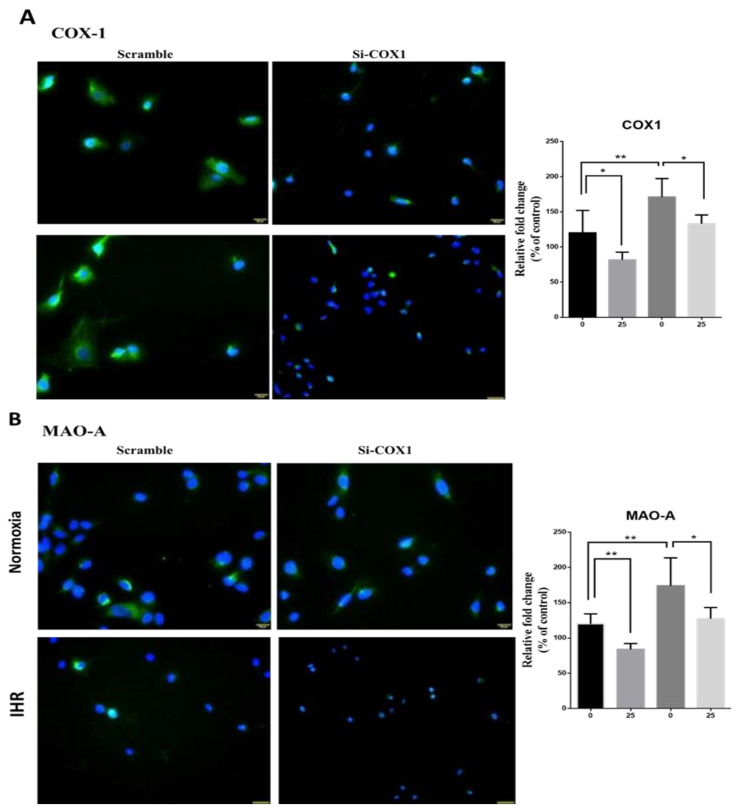
PTGS1 (COX1) knock-down reversed intermittent hypoxia with re-oxygenation (IHR)-induced up-regulation of MAOA hyperactivity. Representative micrographs of immunofluorescence staining in SH-SY5Y neuron cells with or without the knock-down of *PTGS1* under IHR versus normoxic (NOX) conditions are given for (**A**) COX1, and (**B**) MAOA. IHR resulted in over-expressions of COX1, and MAOA hyperactivity, both of which were reversed with the knock-down of COX1. DAPI (blue) is used for staining the nuclei. Localizations of the two molecules are indicated in green. All the micrographs are a merge of the two stainings. Quantified values are stratified based on the response to IHR stimuli and *PTGS1* SiRNA transfection. Kruskal–Wallis test with post-hoc analysis was used for comparisons between four groups. * *p* < 0.05. ** *p* < 0.01.

**Figure 11 antioxidants-10-01854-f011:**
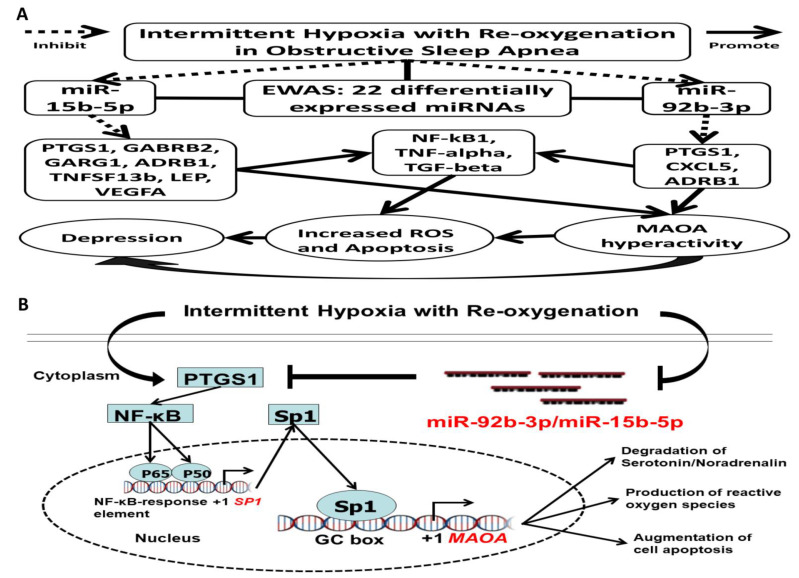
The proposed interplay between miR-15b-5p/miR-92b-3p and their target genes in the intermittent hypoxia with re-oxygenation (IHR) model of obstructive sleep apnea (OSA). (**A**) These interactions contribute to the development of depression by IHR-induced MAOA hyperactivity and IHR-induced oxidative stress/apoptosis/inflammation in OSA patients. (**B**) A schematic diaphragm depicts the protective effect of miR-15b-5p/miR-92b-3p mimics on IHR-induced oxidative stress and cell apoptosis through inhibiting MAOA via targeting PTGS1-NF-κB-SP1 signaling in OSA-related depression. IHR augments PTGS1, which increases the NF-κB expression that in turn activates MAOA gene transcription through augmenting the binding of SP1 to the promoter region of the MAOA gene. Down-regulation of miR-15b-5p/miR92b-3p in response to chronic IHR in OSA leads to MAOA hyperactivity, oxidative stress, and cell injury through targeting PTGS1-NF-κB1-SP1 signaling. MAOA = monoamine oxidase A; PTGS1 = prostaglandin-endoperoxide synthase 1; NF-κB1 = nuclear factor kappa B subunit 1; SP1 = Sp1 transcription factor.

**Table 1 antioxidants-10-01854-t001:** Demographic, sleep, and biochemistry data of the discovery and validation cohorts.

	Discovery Cohort			Validation Cohort			
	OSA Patients *N* = 16	Healthy Subjects *N* = 8	*p* Value	Primary Snoring *N* = 20	Treatment-Naïve OSA Patients *N* = 45	OSA Patients on CPAP *N* = 13	*p* Value
Age, years	55.2 ± 14.8	47.9 ± 13.6	0.242	44.6 ± 12.6	48.5 ± 120.9	50.6 ± 7.5	0.248
Male Sex, n (%)	12 (75)	8 (100)	0.121	12 (60)	34 (75.6)	11 (84.6)	0.252
BMI, kg/m^2^	27.3 ± 3.7	25.0 ± 3.2	0.144	25.1 ± 3.6	26.4 ± 3.4	27.3 ± 2.9	0.173
AHI, events/hour	44.5 ± 24.5	NA		2.7 ± 1.8	54.8 ± 19.2	62.5 ± 23.3	<0.001
ODI, events/hour	25.8 ± 27.5	NA		0.8 ± 0.2	41.9 ± 24.7	44.9 ± 29.4	<0.001
Mean SaO2, %	95.0 ± 2.4	NA		96.1 ± 1.6	93.7 ± 2.5	93.7 ± 2.5	0.001
Nadir SaO2, %	78.5 ± 13.2	NA		89.1 ± 3.7	70.4 ± 15.8	72.5 ± 13	<0.001
Snoring index, counts/hour	359.3 ± 278.4	NA		129.7 ± 147.6	374.7 ± 200.1	315.2 ± 189.5	<0.001
Smoking, n (%)	7 (43.8)	3 (37.5)	0.77	4 (20)	11 (24.4)	4 (30.8)	0.78
Alcoholism, n (%)	0 (0)	0 (0)	1.0	0 (0)	0 (0)	1 (7.7)	0.138
Cholesterol, mg/dl	212.1 ± 58.6	183.2 ± 41.5	0.346	186.7 ± 41.8	200.1 ± 38.6	188.4 ± 38.6	0.375
Triglycerides, mg/dl	158 ± 89.8	75.6 ± 29.1	0.071	118.7 ± 71.4	170.1 ± 82.6	143.8 ± 66.6	0.063
Hypertension, n (%)	4 (25)	2 (25)	1.0	3 (15)	17 (37.8)	8 (61.5)	0.023
Diabetes mellitus, n (%)	1 (6.3)	1 (12.5)	0.602	1 (5)	7 (15.6)	1 (7.7)	0.419
Heart disease, n (%)	2 (12.5)	0 (0)	0.296	2 (10)	2 (4.4)	1 (7.7)	0.686
Stroke, n (%)	1 (6.3)	1 (12.5)	0.602	1 (5)	0 (0)	0 (0)	0.23
COPD, n (%)	2 (12.5)	1 (12.5)	1.0	2 (10)	2 (4.4)	2 (15.4)	0.386
CKD, n (%)	0 (0)	0 (0)	1.0	1 (5)	0 (0)	0 (0)	0.23
Depression, n (%)	6 (37.5)	1 (12.5)	0.204	8 (40)	12 (26.7)	2 (15.4)	0.298

BMI = body mass index; AHI = apnea hypopnea index; ODI = oxygen desaturation index; SaO2 = oxygen saturation; COPD= chronic obstructive pulmonary disease; CKD = chronic kidney disease.

**Table 2 antioxidants-10-01854-t002:** Enriched Kyoto Encyclopedia of Genes and Genomes (KEGG) pathways for predicted target genes of the 22 differentially expressed miRNA.

ID	Description	Gene Ratio *	Bg Ratio #	Adjusted *p* Value	q Value	Excluded miRNA
hsa04218	Cellular senescence	85/1996	156/8105	1.19 × 10^−13^	7.06 × 10^−14^	hsa-miR-29c-5p
hsa04390	Hippo signaling pathway	83/1996	157/8105	1.77 × 10^−12^	1.05 × 10^−12^	hsa-miR-29c-5p
hsa04110	Cell cycle	69/1996	124/8105	6.62 × 10^−12^	3.94 × 10^−12^	22 miRs involved
hsa04520	Adherens junction	44/1996	71/8105	1.27 × 10^−9^	7.53 × 10^−10^	hsa-miR-29c-5p
hsa04933	AGE-RAGE signaling pathway in diabetic complications	54/1996	100/8105	1.06 × 10^−8^	6.28 × 10^−9^	22 miRs involved
hsa04150	mTOR signaling pathway	74/1996	155/8105	1.06 × 10^−8^	6.28 × 10^−9^	hsa-miR-29c-5p
hsa04350	TGF-beta signaling pathway	51/1996	94/8105	1.6 × 10^−8^	9.49 × 10^−9^	22 miRs involved
hsa04141	Protein processing in endoplasmic reticulum	78/1996	171/8105	2.97 × 10^−8^	1.76 × 10^−8^	hsa-miR-29c-5p
hsa04115	p53 signaling pathway	42/1996	73/8105	3.89 × 10^−8^	2.31 × 10^−8^	hsa-miR-29c-5p
hsa04510	Focal adhesion	87/1996	201/8105	6.84 × 10^−8^	4.06 × 10^−8^	hsa-miR-29c-5p
hsa04068	FoxO signaling pathway	63/1996	131/8105	7.04 × 10^−8^	4.18 × 10^−8^	hsa-miR-29c-5p
hsa04550	Signaling pathways regulating pluripotency of stem cells	67/1996	143/8105	8.35 × 10^−8^	4.96 × 10^−8^	22 miRs involved
hsa04120	Ubiquitin mediated proteolysis	64/1996	140/8105	4.24 × 10^−7^	2.52 × 10^−7^	hsa-miR-29c-5p
hsa01522	Endocrine resistance	49/1996	98/8105	4.95 × 10^−7^	2.94 × 10^−7^	hsa-miR-29c-5p
hsa04010	MAPK signaling pathway	112/1996	294/8105	1.37 × 10^−6^	8.13 × 10^−7^	22 miRs involved
hsa04668	TNF signaling pathway	53/1996	112/8105	1.38 × 10^−6^	8.17 × 10^−7^	hsa-miR-29c-5p, has-miR-4433b-3p, hsa-miR-574-3p
hsa04152	AMPK signaling pathway	55/1996	120/8105	2.85 × 10^−6^	1.69 × 10^−6^	hsa-miR-574-3p
hsa04810	Regulation of actin cytoskeleton	87/1996	218/8105	3.01 × 10^−6^	1.79 × 10^−6^	22 miRs involved
hsa04211	Longevity regulating pathway	44/1996	89/8105	3.01 × 10^−6^	1.79 × 10^−6^	hsa-miR-29c-5p, has-miR-4433b-3p, hsa-miR-574-3p
hsa04151	PI3K-Akt signaling pathway	128/1996	354/8105	4.15 × 10^−6^	2.47 × 10^−6^	22 miRs involved
hsa05202	Transcriptional mis-regulation in cancer	78/1996	192/8105	4.62 × 10^−6^	2.74 × 10^−6^	hsa-miR-29c-5p, hsa-miR-574-3p
hsa04722	Neurotrophin signaling pathway	53/1996	119/8105	1.01 × 10^−5^	6.01 × 10^−6^	22 miRs involved
hsa05417	Lipid and atherosclerosis	83/1996	215/8105	2.01 × 10^−5^	1.19 × 10^−5^	hsa-miR-29c-5p
hsa04071	Sphingolipid signaling pathway	52/1996	119/8105	2.38 × 10^−5^	1.42 × 10^−5^	hsa-miR-4433b-3p
hsa04530	Tight junction	68/1996	169/8105	2.9 × 10^−5^	1.72 × 10^−5^	hsa-miR-29c-5p
hsa04919	Thyroid hormone signaling pathway	52/1996	121/8105	4.14 × 10^−5^	2.46 × 10^−5^	22 miRs involved
hsa04144	Endocytosis	92/1996	252/8105	7.94 × 10^−5^	4.72 × 10^−5^	22 miRs involved
hsa04710	Circadian rhythm	19/1996	31/8105	8.95 × 10^−5^	5.32 × 10^−5^	hsa-miR-223-5p, hsa-miR-29c-5p, hsa-miR-574-3p
hsa04310	Wnt signaling pathway	65/1996	166/8105	1.13 × 10^−4^	6.69 × 10^−5^	hsa-miR-29c-5p
hsa04910	Insulin signaling pathway	55/1996	137/8105	2.02 × 10^−4^	1.2 × 10^−4^	hsa-miR-29c-5p, hsa-miR-574-3p
hsa05010	Alzheimer disease	124/1996	369/8105	2.32 × 10^−4^	1.38 × 10^−4^	22 miRs involved
hsa04210	Apoptosis	54/1996	136/8105	3.08 × 10^−4^	1.83 × 10^−4^	hsa-miR-29c-5p, hsa-miR-574-3p
hsa04066	HIF-1 signaling pathway	45/1996	109/8105	4.05 × 10^−4^	2.4 × 10^−4^	22 miRs involved
hsa04340	Hedgehog signaling pathway	27/1996	56/8105	4.85 × 10^−4^	2.88 × 10^−4^	hsa-miR-29c-5p, hsa-miR-574-3p
hsa04015	Rap1 signaling pathway	75/1996	210/8105	8.14 × 10^−4^	4.84 × 10^−4^	hsa-miR-4433b-3p
hsa04935	Growth hormone synthesis, secretion and action	47/1996	119/8105	9.53 × 10^−4^	5.66 × 10^−4^	22 miRs involved
hsa04657	IL-17 signaling pathway	39/1996	94/8105	9.53 × 10^−4^	5.66 × 10^−4^	hsa-miR-223-5p, hsa-miR-29c-5p, hsa-miR-574-3p
hsa04962	Vasopressin-regulated water reabsorption	21/1996	44/8105	2.797 × 10^−3^	1.662 × 10^−3^	hsa-miR-223-5p, hsa-miR-4433b-3p

* Gene ratio stands for the number of genes involved in this pathway among miRNA target genes over the number of miRNA target genes. # Bg ratio stands for the number of all genes of this pathway over the number of the genes of all pathways in Gene Ontology (biological function).

## Data Availability

All the sequencing data are publicly available at the NCBI website with an accession number of GSE174245. The other data presented in this study are available in the article or Appendix A.

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
