# Peer review of "MicroRNA Sequencing Analysis in Obstructive Sleep Apnea and Depression: Anti-Oxidant and MAOA-Inhibiting Effects of miR-15b-5p and miR-92b-3p through Targeting PTGS1-NF-κB-SP1 Signaling"

_antioxidants, 2021, doi:10.3390/antiox10111854_

Round 1

Reviewer 1 Report

Although the topic of the reviewed manuscript is very interesting, unfortunately, it was prepared poorly and made the text difficult to understand. 
The authors have to correct figures 4-8 as with are entirely illegible. The amount of data in the individual figures is too large.
Figure 8 provides only a preliminary study. The authors should examine the individual molecular pathways and present them in detail to prove their assumptions.
Figure 9 is poorly prepared. Authors should present high-resolution microphotography, perhaps fragments of them, to confirm their hypothesis.
Unfortunately, the paper is not suitable for publication in Antyoxydant because the presentation is too sloppy to support the presented conclusions.
I also ask the authors to correct minor but challenging to read typos or bypass superscripts and subscripts.

Author Response

Thank you for your comments. As suggest, we prepare the manuscript again in a legible manner, reduce the amount of data in individual figure, correct the typo errors, and provide high-resolution micrographs. We also acknowledge the limitation that this is only an initial study report in the discussion section as follows:

“Because this report is an initial study on the role of miR-15b/miR-92b in MAOA activity for the development of OSA-related depression, a limitation should be acknowledged. The underlying mechanism of PTGS1, the miR-15b/miR-92b common targeting protein, involved in the pathogenesis of depression, is still ambiguous. Notwithstanding that limitation, this study does correlate miR-15b/miR-92b under-expression and PTGS1 over-expression with MAOA hyperactivity, oxidative stress, and augmented apoptosis of neuron cells in OSA-related depression. Our study sheds considerable light on the direction for developing a therapeutic approach for alleviating OSA-related depression and provides a potential target.”

Reviewer 2 Report

The manuscript entitled “ microRNA Sequencing Analysis in Obstructive Sleep Apnea 2 and Depression: Anti-oxidant and MAOA-Inhibiting Effects of 3 miR-15b-5p and miR-92b-3p through Targeting PTGS1-NF-κB-4 SP1 Signaling” describes OSA-associated miRNAs and suggests a new therapeutic strategy for OSA-related depression using 2 miRNA pathways.

It would be more plausible if authors provide more details of miR-15b-5p/miR-92b-3p about the functional roles and possible relevance to the pathophysiology of depression in the discussion section.

Author Response

Thank you for your comments. As suggest, we provide more details of miR-15b-5p/miR-92b-3p about the functional roles and possible relevance to the pathophysiology of depression in the discussion section as follows: “Our study examined the previously untouched area of the effect of miR-15b/miR-92b on OSA-related depression in vivo and in vitro. Clinical data showed that both miR-15b-5p and miR-92b-3p were down-regulated in OSA patients, while their common target gene, PTGS1, was up-regulated, particularly in those with depression. In vitro experiments showed that miR-15b/miR-92b targeting of PTGS1-NF-κB1-SP1 signaling selectively inhibit MAOA activity. To validate the result, we confirmed a reduction of MAOA activity and ROS production by using immunofluorescence method with the knock-down of PTGS1 under normoxic and IHR conditions. Our findings indicate that miR-15b-5p/miR-92b-3p mimics may alleviate the neuronal damage and MAOA hyperactivity caused by chronic IHR via inhibiting neuroinflammation and oxidative stress. Because this report is an initial study on the role of miR-15b/miR-92b in MAOA activity for the development of OSA-related depression, a limitation should be acknowledged. The underlying mechanism of PTGS1, the miR-15b/miR-92b common targeting protein, involved in the pathogenesis of depression, is still ambiguous. Not-withstanding that limitation, this study does correlate miR-15b/miR-92b under-expression and PTGS1 over-expression with MAOA hyperactivity, oxidative stress, and augmented apoptosis of neuron cells in OSA-related depression. Our study sheds considerable light on the direction for developing a therapeutic approach for alleviating OSA-related depression and provides a potential target.”

Round 2

Reviewer 1 Report

Unfortunately, the authors did not respond to my essential requests. Figure 4, Figure 6, and Figure 7 are still not legible.
Figure 8 and Figure 9 (the first one in the reviewed manuscript) - still have illegible captions under the charts.
Figure 9 (currently 10, no correction in the text) is still poorly prepared. As I previously suggested, authors should present high-resolution microphotography, perhaps fragments of them, to confirm their hypothesis.
The authors added a description:
Because this report is an initial study on the role of miR-15b / miR-92b in MAOA activity for the development of OSA-related depression, a limitation should be acknowledged. The underlying mechanism of PTGS1, the miR-15b / miR-92b common targeting protein, involved in the pathogenesis of depression, is still ambiguous. Notwithstanding that limitation, this study does correlate miR-15b / miR-92b under-expression and PTGS1 over-expression with MAOA hyperactivity, oxidative stress, and augmented apoptosis of neuron cells in OSA-related depression. Our study sheds considerable light on the direction for developing a therapeutic approach for alleviating OSA-related depression and provides a potential target. "
Unfortunately, the information does not enrich the manuscript but only highlights its most critical shortcomings - the lack of the study of molecular pathways involved in the observed modulation.

Author Response

Response to Reviewer (round 2)

Unfortunately, the authors did not respond to my essential requests. Figure 4, Figure 6, and Figure 7 are still not legible.

Figure 8 and Figure 9 (the first one in the reviewed manuscript) - still have illegible captions under the charts.

Figure 9 (currently 10, no correction in the text) is still poorly prepared. As I previously suggested, authors should present high-resolution microphotography, perhaps fragments of them, to confirm their hypothesis.

The authors added a description:

Because this report is an initial study on the role of miR-15b / miR-92b in MAOA activity for the development of OSA-related depression, a limitation should be acknowledged. The underlying mechanism of PTGS1, the miR-15b / miR-92b common targeting protein, involved in the pathogenesis of depression, is still ambiguous. Notwithstanding that limitation, this study does correlate miR-15b / miR-92b under-expression and PTGS1 over-expression with MAOA hyperactivity, oxidative stress, and augmented apoptosis of neuron cells in OSA-related depression. Our study sheds considerable light on the direction for developing a therapeutic approach for alleviating OSA-related depression and provides a potential target. "

Unfortunately, the information does not enrich the manuscript but only highlights its most critical shortcomings - the lack of the study of molecular pathways involved in the observed modulation.

Ans.: Thanks again for your comments. As suggest, we provide Figures 6, 7, 8, & 9 with clear captions. We also provide Figure 10 with distinct micrographs in five fragments, and three of them are moved to Supplementary material. As to Figure 4, the KEGG pathways plots were generated with the KEGG website and then converted to high resolution image files. Because of their copyright claims, it is impossible to increase the font size or thicken the lines. However, we have tried our best to improve the resolution by using graphics software. Hopefully, these amendments would make the manuscript more readable and acceptable.

Round 3

Reviewer 1 Report

Sorry for delaying the review, but the attached file with comments is hard to read. I thank the authors for modifications in the figures according to my suggestions. Unfortunately, figure 4 is still challenging to analyze.
The main problem in the manuscript is that the authors still do not consider the more comprehensive study. As  I emphasized from the beginning suggested in the title of the paper, the PTGS1-NF-κB-SP1 pathway should be carefully analyzed to confirm that supposition. Unfortunately, in their current form of the manuscript, the authors presented only preliminary studies, such do not constitute a valuable manuscript.